# Tropical Forest Landscape Restoration in Indonesia: A Review

Yonky Indrajaya [1,*], Tri Wira Yuwati [2], Sri Lestari [3], Bondan Winarno [3], Budi Hadi Narendra [3], Hunggul Yudono Setio Hadi Nugroho [1], Dony Rachmanadi [2], Pratiwi [3], Maman Turjaman [3], Rahardyan Nugroho Adi [1], Endang Savitri [1], Pamungkas Buana Putra [1], Purwanto Budi Santosa [2], Nunung Puji Nugroho [1], Sigit Andy Cahyono [1], Reni Setyo Wahyuningtyas [2], Retno Prayudyaningsih [4], Wawan Halwany [2], Mohamad Siarudin [5], Ary Widiyanto [5], Marcellinus Mandira Budi Utomo [5], Sumardi [6], Aji Winara [7], Tien Wahyuni [8] and Daniel Mendham [9]

[1] Watershed Management Technology Center, Jl. Ahmad Yani, Surakarta 57102, Indonesia;
hunggulys@yahoo.com (H.Y.S.H.N.); dd11lb@yahoo.com (R.N.A.); savitriendang@gmail.com (E.S.);
pamungkas_buanaputra@yahoo.co.id (P.B.P.); npnugroho09@gmail.com (N.P.N.);
sandycahyono@yahoo.com (S.A.C.)
[2] Environment and Forestry Research and Development Institute of Banjarbaru, Jl. Ahmad Yani Km 28.7,
Banjarbaru 70721, Indonesia; yuwatitriwira@gmail.com (T.W.Y.); dony.research@gmail.com (D.R.);
pur.befordi@gmail.com (P.B.S.); renisetyowahyuningtyas@gmail.com (R.S.W.); wawanh73@gmail.com (W.H.)
[3] Forest Research and Development Center, The Ministry of Environment and Forestry, Jl. Gunung Batu No. 5,
Bogor 16118, Indonesia; lestari@iuj.ac.jp (S.L.); bondanw2308@gmail.com (B.W.);
narendra17511@gmail.com (B.H.N.); pratiwi.lala@yahoo.com (P.); turjaman@gmail.com (M.T.)
[4] Environment and Forestry Research and Development Institute of Makassar, Jl. P. Kemerdekaan Km 16,
Makassar 90243, Indonesia; rprayudyaningsih@gmail.com
[5] Agroforestry Research and Development Center (ARDC), The Ministry of Environment and Forestry,
Jalan Raya Ciamis-Banjar Km 4, Ciamis 46271, Indonesia; msiarudin@yahoo.com (M.S.);
ary_301080@yahoo.co.id (A.W.); marcell.utomo@gmail.com (M.M.B.U.)
[6] Center for Forest Biotechnology and Tree Improvement, Jl. Palagan Tentara Pelajar Km 15, Purwobinangun,
Pakem, Yogyakarta 55582, Indonesia; sumardi_184@yahoo.com
[7] Research and Development Agency of West Java Province (BP2D), Jl. Kawaluyaan Indah Raya No. 6,
Bandung 40286, Indonesia; awinara1@gmail.com
[8] Research and Development Centre for Dipterocarps Forest Ecosystem, Jl. A. Wahab Syahranie No.68,
Sempaja Sel., Kec. Samarinda Utara, Kota Samarinda 75119, Indonesia; yunitien@hotmail.com
[9] CSIRO Land and Water, Private Bag 12, Hobart, TAS 7001, Australia; daniel.mendham@csiro.au
* Correspondence: yonky_indrajaya@yahoo.com; Tel.: +62-813-9257-7088

**Abstract:** Indonesia has the second-largest biodiversity of any country in the world. Deforestation and forest degradation have caused a range of environmental issues, including habitat degradation and loss of biodiversity, deterioration of water quality and quantity, air pollution, and increased greenhouse gas emissions that contribute to climate change. Forest restoration at the landscape level has been conducted to balance ecological integrity and human well-being. Forest restoration efforts are also aimed at reducing $CO_2$ emissions and are closely related to Indonesia's Nationally Determined Contribution (NDC) from the forestry sector. The purpose of this paper is to examine the regulatory, institutional, and policy aspects of forest restoration in Indonesia, as well as the implementation of forest restoration activities in the country. The article was written using a synoptic review approach to Forest Landscape Restoration (FLR)-related articles and national experiences. Failures, success stories, and criteria and indicators for forest restoration success are all discussed. We also discuss the latest silvicultural techniques for the success of the forest restoration program. Restoration governance in Indonesia has focused on the wetland ecosystem such as peatlands and mangroves, but due to the severely degraded condition of many forests, the government has by necessity opted for active restoration involving the planting and establishment of livelihood options. The government has adapted its restoration approach from the early focus on ecological restoration to more forest landscape restoration, which recognizes that involving the local community in restoration activities is critical for the success of forest restoration.

**Keywords:** reforestation; land rehabilitation; comprehensive perspective; livelihoods

## 1. Introduction

Indonesia consists of 17,000 islands with a range of habitats and biogeographic, geological, climatic, and ecological areas. This has resulted in high biodiversity and a high number of endemic species [1]. While Indonesia's land area represents 1.3% of the Earth's surface, it contributes substantially more to the world's biodiversity, comprising around 11% of the world's plant species, 10% of mammal species, and 16% of bird species [2]. Forests play an important role in climate and water cycle regulation, the global carbon cycle, and the world's terrestrial biodiversity [3].

As the timber industry has grown, the extent and rate of deforestation have increased significantly. The country's forest cover has decreased from 74% to 56% in the 30–40 years leading up to 1990 [4]. Sunderlin and Resosudarmo [5] reported that the progression of annual deforestation was as follows: In the 1970s, 300,000 ha were deforested annually; in 1981, 600,000 ha were deforested annually; and in 1990, 1 million ha were deforested annually. MoEF [6] reported that the highest levels of deforestation rates were recorded from 1996 to 2000, at 3.51 million hectares per year. From 2002 to 2014, the rate of deforestation declined, along with a decline in the incidences of forest and land fires [6]. After 2015, the deforestation rate decreased to an average of <1 million ha per year, and in 2019, it was 0.46 Mha [6].

Tropical deforestation has far-reaching and long-term environmental consequences [7]. These consequences include habitat degradation and loss of biodiversity, deterioration in water quality and quantity, air pollution, and increased greenhouse gas emissions, contributing to climate change [8,9]. To deal with the degradation and deforestation, forest rehabilitation efforts in Indonesia started in the 1950s, with different approaches depending on the political regime [10,11]. From the 1950s to the 1970s, forest rehabilitation policies were primarily 'top-down' but had become more participative, involving the communities, by the end of the 1990s [10]. Rehabilitation programs were in a state of flux between the 1980s and the mid-1990s. Rehabilitation became a top priority after the Ministry of Forestry (MoF) separated from the Ministry of Agriculture in 1983. Since the reformation era in 1998, the shift from privately owned and large-scale forest management to smaller-scale community-based forest management has gained traction [10].

Recent global policy debates about terrestrial ecosystem restoration have centered on the concept of forest and landscape restoration, which has been defined as "the long-term process of regaining ecological functionality and improving human well-being across deforested or degraded forest landscapes" [12,13]. As with other landscape approaches, forest and landscape restoration aims to balance ecological integrity and human well-being [14] by establishing a mosaic of interdependent land uses such as crop and livestock production, agroforestry, improved fallow systems, mining, ecological corridors, discrete areas of forest and woodland, and riparian plantings that protect watercourses [15,16]. Its objective is to use comprehensive spatial planning to allocate land uses more efficiently and improve their individual and collective sustainability in cooperation with landowners and other stakeholders [9,17]. For example, ecological intensification techniques can boost agricultural productivity in selected landscape areas while allowing natural regeneration to occur elsewhere [18]. Restoration of mixed-use agricultural land between primary forest areas can create wildlife corridors [15].

Since 2010, a number of international initiatives have set lofty goals for forest restoration around the world [19]. Indonesia started implementing land rehabilitation in the 1950s and has been recognized as a pioneer of forest landscape restoration (FLR), a term recognized in the 2010s [20]. Restoration programs in Indonesia were mostly aimed at ecological purposes in forest and nonforest areas. Over time, they have adopted a more multifunctional mosaic landscape approach (FLR) [21]. The forest ecosystem in Indonesia encompasses the forest in lowland tropical rain forest, upland forest, monsoon forest, savanna, peat swamp forest, and mangrove forest [22]. Rehabilitation efforts have been conducted in all of these ecosystems. The government classified rehabilitation efforts as either

reforestation (replanting of forest land areas) in state forests or regreening/afforestation (replanting of nonforest land areas) in community areas outside of state forests [10].

In 2011, the Bonn Challenge was launched with the goal of restoring 350 million hectares (Mha) by 2030 [23]. In its Nationally Determined Contribution (NDC) to the Paris Agreement, Indonesia set a lofty goal of restoring 2 million hectares of peatland and rehabilitating 12 million hectares of degraded land by 2030 [24]. Various restoration projects have shown that it is possible to recover the original composition, function, and formation of natural ecosystems at particular sites [25]. Some argue, nevertheless, that the high costs and long timescales sometimes involved with ecosystem restoration projects mean that certain initiatives cannot achieve restoration at the large spatial scales required to have a significant global impact [26]. Landscape-scale restoration efforts focus on recreating the conditions that make natural regeneration occur, aided where required by judicious planting or other interventions [9]. When resources are scarce, restoration can be carried out in stages, such as restoring forest cover first and then focusing on the broader aspects of ecological complexity needed for resilience against climate change and other pressures [27].

In Indonesia, although restoration efforts have been pursued for several decades—national projects have even been carried out several times—their effectiveness at restoring degraded forests has yet to be fully realized. This happens because social aspects such as conflict resolution and clarity on tenure, low level of community participation due to a project-based orientation, and weak institutional arrangements to establish effective implementation have not been handled well [28]. In addition, the physical implementation of restoration does not consider or adjust to land characteristics, species selection, planting, and maintaining management [10,29]. To support forest restoration activities, the Indonesian government issued Government Regulation 26/2020, which is a derivative of Law 11/2020 on job creation [30]. This regulation notes that forest rehabilitation is important for achieving adequate forest area and the restoration of watershed conditions. Restoration requires more than reforestation: it also needs to be supported by forest maintenance, social forestry, forest protection, soil and water conservation, and forest rehabilitation by community institutional and technological development. Restoration activities should also be planned, implemented, and managed to ensure that the restored forest is profitable for the community and that its ecological functions remain sustainable [31]. This comprehensive perspective is critical for forest landscape restoration since it emphasizes the need to address deforestation and land degradation sources.

This review paper aims to provide an overview of forest landscape restoration progress in Indonesia, including the basis of implementation, supporting and inhibiting factors, as well as the benefits obtained. The article is prepared with a synoptic review approach to FLR-related articles and nationwide experiences (see Figure 1). Materials in the review come from national and international research papers, technical reports, and relevant books. The scope of the review includes theoretical perspectives on FLR based on hydrological, biodiverse, carbon, and socioeconomic factors. It also examines the regulations, institutions, and policy aspects supporting FLR, and discusses the implementation of FLR, including the criteria and indicators and their benefits in climate change mitigation, hydrology, soil improvement, biodiversity, and livelihood. The review also involves applied silvicultural techniques, monitoring, and evaluation.

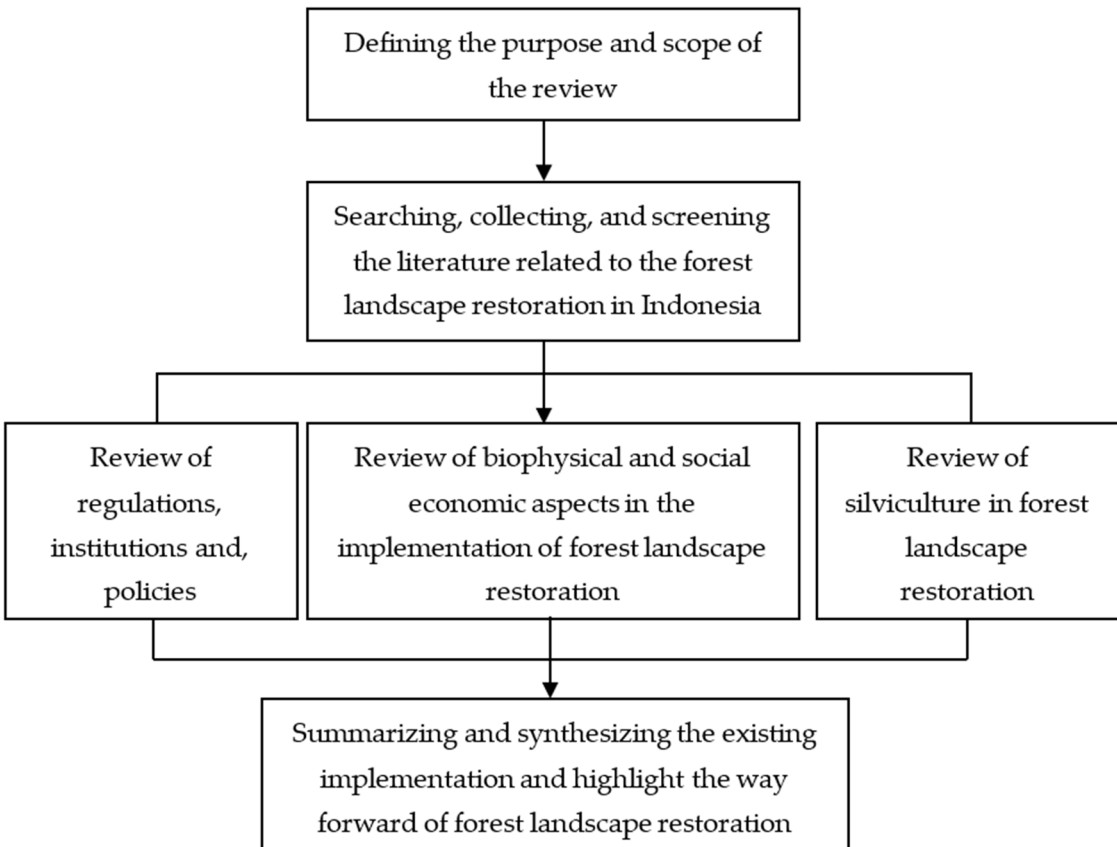

**Figure 1.** Stages in conducting the review.

## 2. Theoretical Perspectives on Land and Forest Restoration

Although the overall conceptual framework of forest restoration at the landscape level is relatively new, understanding the principles and techniques behind the approach has existed for a long time and is already familiar to many forestry practitioners [12]. Common terms previously used, such as rehabilitation and reforestation as conventional restoration, have been around for decades. Landscapes offer additional ecosystem services such as food and water, climate and disease regulation, nutrient cycling, and plant pollination, as well as recreational benefits. Furthermore, this section will describe how important FLR is from the perspective of natural science (hydrology, biodiversity, and carbon storage) and socioeconomic considerations.

### 2.1. Hydrology, Biodiversity, and Carbon

It is widely recognized that forests play an important role in intercepting rain, decreasing surface runoff velocity on slopes, reducing erosion [32], and increasing infiltration opportunities [33]. Indonesia's natural forests are also home to extensive biodiversity, ranging from the evergreen lowland forests in Sumatra and Kalimantan to seasonal forests and savanna grasslands in Nusa Tenggara and mountainous areas in Papua [34]. Forest ecosystems are also important because they are the largest carbon sink in terrestrial ecosystems [35] and strongly contribute to climate change mitigation by absorbing atmospheric $CO_2$.

Deforestation and forest degradation have continued to occur over the last two decades, resulting in a reduction in the forest's hydrological function and alteration of the local hydroclimate, now characterized by increased runoff, increased soil erosion, and decreased base flow [36–38]. This situation increases the risk of flooding, which is already exacerbated by the more frequent extreme rainfall due to climate change [39]. Deforestation in Indonesia also endangers many native species. Based on the IUCN Red List for

Threatened Species, 3393 plants and 674 endemic species are almost extinct, and their risk of extinction will continue to increase as Indonesia's forest cover continues to decline [40]. Deforestation and degradation also result in a decrease in carbon stocks. For example, the conversion of primary forest to secondary forest potentially reduces the carbon stock by 80% in dryland forest, 89% in peat swamp forest, and 71% in mangrove forest [41]. Forests absorb large amounts of carbon dioxide ($CO_2$) from the atmosphere and store it in living and dead biomass and soil [42]. Carbon storage in forest ecosystems is 20–100 times greater per unit area than in agricultural land [43]. Mangroves can absorb an average of 892 tons/ha of carbon [44]. Restoration of tropical peatlands is also important.

Forest landscape restoration is critical for the restoration of hydrological, biodiverse, and carbon functions, especially under a changing climate and the anticipated extreme weather that is forecast to occur more frequently in the future [45]. Successful restoration also plays a role in improving soil characteristics on the forest floor by increasing soil organism activity, root development, and organic matter content [32]. The restoration that results in an outcome similar to natural forest conditions can result in total biomass of coarse and fine roots that is double that of the same aboveground biomass in plantations, with flow-on effects on soil organic matter [46]. Successful forest landscape restoration also supports conservation of biodiversity and can help mitigate climate change.

*2.2. Socioeconomic Considerations*

There are relatively few studies worldwide on the socioeconomic aspects of restoration when compared to those based on ecological aspects, and they tend to focus on specific issues such as local community engagement, resource investment, job and income generation [25,47,48], and psychological outcomes [49]. Economic research on forest cover change has tended to focus on deforestation rather than restoration [50,51]. Even evaluations of the benefits and costs of forest restoration programs are scarce and incorrect [52]. To be attractive to local communities, forest landscape restoration must deliver socioeconomic benefits. Local people must obtain benefits from reforestation that outweigh those from other land uses; otherwise, there is a significant disincentive to reforestation [53]. Local communities require revenue to survive and incentives to reduce their preferences for livelihood options that do not support the restoration activities at the field level [54,55]. Encouraging people to switch to new livelihoods is challenging due to the scarcity of alternatives that are competitive with existing livelihoods [56]. Additional funding is critical to sustaining the restoration effort [57] and expanding viable livelihood possibilities.

Restoration requires adequate funding, so economic concepts are critical for restoring biodiversity and ecosystem processes [58]. However, such economic concepts are often flawed because they undervalue the ecosystem services provided for future generations because future values are frequently externalized and discounted, often diminishing exponentially into the future [59,60]. Even with sound planning in place, the financial requirement for restoration often exceeds the available resources. The success of future restoration projects depends on economic efficiency and appropriate levels of investment [58,61]. Social cost-benefit analysis has proven to be an effective instrument for restoration [62].

Forest landscape restoration is carried out to achieve a balance of biophysical and socioeconomic benefits. Efforts to maximize financial returns from forest products or ecosystem services will likely sacrifice other biophysical and/or social benefits [17]. Recognizing the public benefits of reforestation, land managers will restore their land with a more certain future profit/private good orientation. As the state forest manager, the government must plan the restoration of its forests in a landscape context, be prepared for long-term investment, and consider the interaction between the community and the area to be restored.

**3. Regulation, Institutions, and Policies of Forest Restoration Governance**

The framework for effective natural resource management is set by governance [63]. Governance defines who makes decisions and how those decisions are implemented. Sig-

nificant shifts in natural resource governance have occurred in recent decades: while until the 1970s, governance was synonymous with the 'command and control' nature of central governments, trends indicate a shift towards decentralization and the expanding role of both civil society and the private sector in natural resource governance [64]. Forest landscape restoration governance in Indonesia started with different parties with different goals [21]. However, all the forest landscape governance activities demonstrate a dynamic process of bringing the landscape's stakeholders together and a flexible process of creating institutional space for conflict resolution, negotiation, and decision-making at the landscape level. As a result, the focus of the landscape shifted from reserved forests to forested mosaic lands over time. Here we discuss the Indonesian government's regulations, institutions, and policies for reducing the degradation and loss of forest functions and restoring their condition.

*3.1. Forest Restoration Laws and Regulations*

Since Indonesia's independence, there have been three eras of government that have fundamentally influenced the legal system, including forestry laws, namely the old order regime/Soekarno's era (1945–1966), the new order regime/Soeharto's era (1966–1998), and the reformation regime (post-1998). The three eras have their own characteristics and perspectives in relation to forestry issues and have produced different types of forest laws and policies [65]. There was a decentralized policy supported by socialism and anti-Western nationalism in the old order [66], followed by a prowestern centralized era [66,67], and decentralization again in the reformation era. An examination of existing laws and regulations revealed that there is not a single law or regulation related to forestry and the environment that literally/textually contains the phrase "restorasi hutan" ("forest restoration"), but forest restoration is covered in several forestry-related laws and regulations in the form of "restorasi ekosistem" (ecosystem restoration), "restorasi gambut" (peat restoration), "restorasi hidrologi" (hydrology restoration), and "restorasi habitat" (habitat restoration).

To analyze which regulations conceptually contain forest restoration and what forest-related activities are considered restoration activities, it is first necessary to understand the broad definition of forest restoration. Broad definitions of restoration include:

- WWF: Forest restoration is "the process of improving the health, productivity, and array of life of a forest" [68].
- WWF and IUCN: Forest landscape restoration is "a planned process that aims to regain ecological integrity and enhance human well-being in [on] deforested or degraded forest landscapes [and beyond]." In general, it aims at a progression toward higher forest quality from the perspectives of both ecological integrity and human well-being at a landscape scale [69]
- Forest landscape restoration refers to the restoration of forest ecological services within the landscape unit: not returning it to its original state, but restoring its function in terms of biodiversity protection, ecological function, and community livelihoods and incomes [21].
- Forest restoration is more than just planting trees. It is about reinstating the balance of the ecological, social, and economic benefits of forests and trees within a broader pattern of land use [70].

Referring to these broad definitions of restoration, MoEF implemented several restoration-related activities, including peat restoration, ecosystem restoration, forest and land rehabilitation and reclamation, ex-mining area reclamation, afforestation and reforestation, and forest restoration. Thus, based on the broad definition of forest restoration, the Indonesian government has formally regulated it since independence through several levels of regulations. The laws and regulations and the context of forest restoration they govern are presented in Table 1.

**Table 1.** Laws and regulations related to forestry in Indonesia governing forest restoration.

| No. | Regulation | Restoration-Related Activities |
|-----|------------|-------------------------------|
| **A** | **Laws** | |
| 1 | UU 5 of 1967 on Forestry Main Provision | Afforestation |
| 2 | UU 41 of 1999 on Forestry | - Gradual rehabilitation of forest and land in an effort to restore and develop the function of forest and land resources, both production functions and protection and conservation functions to restore, maintain, and improve forest and land functions so that their carrying capacity, productivity, and role in supporting life is maintained<br>- Forest reclamation to repair or restore damaged land and forest vegetation so that they can function optimally according to their designation<br>- Restoration of forest conditions |
| 3 | UU 37 of 2014 on Soil and Water Conservation | - Forest and land rehabilitation<br>- Restoration of soil function on critical land in protected areas (peat areas, protected forests, catchment areas, river borders, green open spaces, around lakes, reservoirs, and springs) |
| | UU 18 of 2013 on prevention and eradication of forest destruction | - Obligation to restore forest condition<br>- Government coercion as a legal action so that companies/legal entities carry out forest restoration due to their actions of destroying forests because they do not comply with the provisions of the legislation |
| **B** | **Government Regulation** | |
| | PP 76 of 2008 on Forest Rehabilitation and Reclamation, which was later changed to PP 26 of 2020 | Rehabilitation and reclamation of forests to restore, maintain, and improve the functions of forests and lands in order to increase their carrying capacity, productivity, and their role supporting life |
| **C** | **Presidential Decree/Instruction** | |
| | Presidential Instruction on afforestation and reforestation assistance (started in 1976–1982) | Afforestation and reforestation for each fiscal year in areas that are urgent, especially in critical areas in watersheds |
| **D** | **Ministry Regulations** | |
| 1 | P.02/Menhut-V/2004 on the procedures for implementing the National Movement for Forest and Land Rehabilitation. | Forest and Land Rehabilitation Movement |
| 2 | P.105/MENLHK/SETJEN/KUM.1/12/2018 on Implementation Procedures, Support Activities, Providing Incentives, as well as Coaching and Control Forest and Land Rehabilitation Activities, which were later replaced with P.2/MENLHK/SETJEN/KUM.1/1/2020 | Forest and Land Rehabilitation |

*3.2. Institutions Involved in Forest Restoration*

In relation to forest restoration in Indonesia, the only formal institution that uses the terminology of restoration is the Peat and Mangrove Restoration Agency. This institution was established in 2016, through Presidential Regulation No. 1 of 2016 concerning the establishment of the "Badan Restorasi Gambut" or Peatland Restoration Agency (PRA) to coordinate and facilitate the restoration of 2 million hectares of degraded peatland in seven provinces (Riau, Jambi, South Sumatra, West Kalimantan, Central Kalimantan, South Kalimantan, and Papua) [71]. In 2020, through Presidential Decree No. 120 of 2020, this agency was changed to the "Badan Restorasi Gambut dan Mangrove" or Peat and

Mangrove Restoration Agency (PMRA) and received an additional mandate to restore mangrove areas in nine provinces (North Sumatra, Riau, Riau Islands, Bangka Belitung, West Kalimantan, East Kalimantan, North Kalimantan, Papua, and West Papua) [72]. With the support of the MoEF and the provincial government, four laws, and one presidential regulation, these institutions have significant influence over the success of peat and mangrove restoration [73]. The institution in the MoEF assigned to peat restoration is the Directorate of Peat Damage Control, under the Directorate General of Pollution Control and Environmental Damage.

There are also several institutions, at the level of directorate general under MoEF, that have a mandate and the authority to restore forest ecosystems [74]. In production forest areas, the Directorate General of Sustainable Forest Management (DGSFM) is responsible for forest areas managed by third parties through the mechanism of forest product utilization permits and borrow-to-use forest area permits. Ecosystem restoration in nature reserve areas and nature conservation areas is the responsibility of the Directorate General of Natural Resources and Ecosystem Conservation (DGNREC) with multiparty implementers, including third parties, through rehabilitation and restoration implementation permits by business entities [75]. The Directorate General of Social Forestry and Environmental Partnerships (DGSFEP) focuses on forests allocated for social forestry stipulated in an indicative map of social forestry area [76]. Devolvement of forest management to social forestry permit holders is deemed a positive mechanism to achieve forest restoration [77,78]. The last institution, the Directorate General of Watershed Management-Forest Rehabilitation (DGWM-FR) is responsible for critical land inside and outside forest areas based on national critical land maps [79]. The role of the Ministry of Environment and Forestry is vital in forest restoration [80], from the planning stage through to monitoring forestry activities in state forest areas [81,82].

The MoEF devolves forest management authority to the provincial governments through the forestry service and forest management unit (FMU). The establishment of FMUs is an effort to strengthen national and provincial forest management systems [81]. Specifically for the provinces on the islands of Java and Madura, the authority for forest restoration in protected forest areas and production forests is given to a state-owned enterprise, namely "Perum Perhutani" [83]. Table 2 and Figure 1 show the stakeholders involved in Indonesia's forest restoration.

**Table 2.** Key stakeholders in Indonesian forest restoration.

| Stakeholder | Type of Activity | Interest | Power | Involvement |
|---|---|---|---|---|
| MoEF | Mandatory | High | High | Direct |
| PMRA | Mandatory | High | High | Direct |
| Permit holders (private sector) | Mandatory | High | Low | Direct |
| Provincial forestry service | Mandatory | High | Low–High | Direct |
| Forest Management Unit and forestry state-owned enterprise | Mandatory | High | Low–High | Direct |
| Civil society organizations (CSOs), Nongovernmental organizations (NGOs), academia and environmental activists, philanthropic, and international development agencies | Voluntary | High | Low–Moderate | Indirect– Direct |
| Other ministries | Ad Hoc | Low | Low–High | Indirect–Direct |

Table 3 shows that the key players in forest restoration are the MoEF, the PMRA, the FMU, the provincial forestry service (under governor supervision), and the forestry state-owned enterprises (in Java and Madura). Other stakeholders, especially governmental bodies, are sometimes prevented from contributing actively to forest restoration by limited funding, and their power does not effectively influence restoration policy and implementation [84]. Limited funding is a common obstacle in landscape restoration in the tropics [85]. Other than governmental bodies, the role of CSOs and NGOs in restoration/afforestation is

crucial, which is common for low-income nations [86]. CSOs and NGOs can help connect restoration efforts with philanthropic and international development agencies to finance programs [87]. Universities/academics and banking institutions, both individually and collectively, contribute to the forest landscape restoration in Indonesia [78].

**Table 3.** Power-interest matrix of forest restoration in Indonesia.

| High | **Subjects** | | **Players** |
|---|---|---|---|
| | • Permit holders: logging concession, mining concession, social forestry permit holder, ecosystem restoration permit holder<br>• CSO/NGO, academia, bank, consortium, environmental activists and other nonforestry private sector players<br>• Enthusiastic community, philanthropic, and international development agencies | • | MoEF, PMRA, FMU, Provincial Forestry Service (under governor supervision), Forestry State-Owned Enterprise |
| **INTEREST** | **Crowd** | | **Context Setters** |
| | • Skeptical community | • | Other ministries: Ministry of Agriculture, Ministry of Marine Affairs and Fisheries, Ministry of Finance, Ministry of National Development Planning, Coordinating Ministry for Maritime and Investment Affairs |
| Low | | | |
| | **Low** | **POWER** | **High** |

Forest landscape restoration is a program that must be mainstreamed and strengthened through spatial planning and law enforcement [88]. Lessons can be taken from Brazil, Guatemala, Colombia, Costa Rica, Ecuador, and Mexico in terms of forest landscape restoration programs that also have a strong legal framework for forest restoration. There are also social-sectoral conflicts in their implementation [89]. However, with adaptive governance that facilitates multistakeholder and multisector local needs, forest restoration can be achieved [90].

*3.3. Forest Restoration Policy*

In terms of forest restoration policies, the establishment of the PMRA was Indonesia's key strategic step in contributing to global environmental improvement. In its implementation, PMRA activities are based on the Provincial Peat Ecosystem Protection and Management Plan (RPPEG) and/or the Provincial Peat Ecosystem Restoration Plan Map. If the Provincial RPPEG or Provincial Peat Ecosystem Restoration Plan Map is not yet available, the implementation of the peat ecosystem restoration is carried out based on the National RPPEG. The RPPEG aims to consider the diversity of ecological characters and functions, natural resource potential, population distribution, local wisdom, community aspirations, climate change, and spatial planning to preserve the peat ecosystem.

Restoration activities in the production forest are the responsibility and authority of the Directorate General of Sustainable Production Forest Management (DGSFM). The Production Forest Utilization Agency, an institution under DGSFM, monitors and evaluates this activity at the regional level. Restoration of production forests aims to restore degraded forest areas, especially in former forest concession (Hak Pengusahaan Hutan/HPH) areas and in industrial plantation forests (Hutan Tanaman Industri/HTI).

Based on the Ministry of Forestry Regulation No. 159 of 2004 on ecosystem restoration in production forests, ecosystem restoration is carried out to restore biotic elements (flora and fauna) and abiotic elements (soil, climate, and topography) in an area so that biological balance is achieved through planting, enrichment, natural regeneration and/or ecosystem protection. The main requirement for ecosystem restoration is that the forest area can no longer be used for production because the canopy cover is less than 60%, and the area is an important habitat refuge for the surrounding ecosystem. This regulation states that the

ecosystem restoration permit holder in a production forest area is prohibited from using trees and/or parts of trees in the area being restored.

Another regulation supporting ecosystem restoration in production forest areas is Ministry of Forestry Regulation No. 61 of 2008 concerning provisions and procedures for granting business permits for utilization of timber forest products for ecosystem restoration in natural forests in production forests (IUPHHK-RE) through applications [91]. This regulation stipulates that the areas that must be restored are ex-HPH areas whose permits have been revoked or whose rights have been returned to the state because the forest area is degraded, or the company does not want to extend it again. The scope of the IUPHHK-RE was then revised through Ministry of Environment and Forestry Regulation No. 28 of 2018 [92] and MoEF Regulation No. 19 of 2019 [93]. The financing of this forest restoration is entirely the responsibility of the permit holder of the IUPHHK-RE. IUPHHK-RE has a concession period of 60 years that can be extended by 35 years. Until 2016 only 16 business units had received permits, with a total restoration concession area of 623,075 ha [94].

For nature reserve areas (NRA) and nature conservation areas (NCA), ecosystem restoration activities are managed through Ministry of Forestry Regulation No. 48 of 2014 concerning procedures for implementing ecosystem restoration in nature reserves and nature conservation areas [95]. The Directorate General of Natural Resources and Ecosystem Conservation (DGNREC) is in charge. NRA or NCA ecosystem restoration plans consists of a long-term ecosystem recovery plan called an ecosystem recovery plan (RPE) and a short-term ecosystem recovery plan called an ecosystem recovery annual activity plan (RKT-PE).

In watershed areas, the responsibility for restoration belongs to the Directorate General of Watershed Management and Forests Rehabilitation (DGWM-FR). This institution has authority to restore, maintain, and improve the function of forests and land so that their carrying capacity, productivity, and role in supporting life systems are maintained. The legal basis is the Ministry of Environment and Forestry Regulation No. 105 of 2018 concerning procedures for implementation, supporting activities, providing incentives, as well as fostering and controlling forest and land rehabilitation activities [79].

The substantial need for forest restoration, together with a strong political will, can form the basis for the government to meet its ambitious commitment to the Paris Agreement. Indonesia's NDC to the Paris Agreement is to reduce greenhouse gas emissions by 29% unconditionally (without international support) and 41% conditionally (with adequate international support) by 2030. Through the active involvement and cooperation of all stakeholders, this commitment has a strong likelihood of being realized, which will be Indonesia's real contribution to global environmental preservation and sustainable development.

## 4. Implementation of Forest Restoration

### 4.1. Progress of Forest Restoration in Indonesia

MoEF [6] reported that reservoirs, priority lake areas, and river basins are rehabilitated, mangrove forests and urban forests are being developed, and community nurseries are being established. Dams and retaining bars, gully plugs, and absorption wells may also be constructed as part of the effort. The total area rehabilitated in 2019 was 395,168 hectares. About 206,000 hectares of conservation and protection forests, 1000 hectares of mangrove forests, beaches, swamps, and peat, and 188,168 hectares of community lands benefitting from community nurseries were included in the total [6]., Table 4 depicts forest and land rehabilitation progress from January 2015 to December 2019. During 2015–2019, 944 check dams and 2330 gully plugs were built. Communities are expected to help with adaptation to disaster relief and climate change, including floods, landslides, and droughts. Considering that the quality of seedlings is critical to the success of land rehabilitation, permanent nurseries were established throughout Indonesia to prepare productive seedlings for planting. In 2019, there were 57 permanent nurseries across the country, with seven of them employing tissue culture technology [6]. Table 4 shows the emergent role of community-based

land rehabilitation (nonforest area) with more multipurpose tree-species (MPTS), which attract and encourage more people to plant trees on their own land.

**Table 4.** Area planted in forest and non-forest area (in ha).

| Type of Area | Year | | | | |
|---|---|---|---|---|---|
| | **2015** | **2016** | **2017** | **2018** | **2019** |
| Conservation/ protection forests | 10,508 | 7067 | 19,482 | 25,170 | 206,000 |
| Mangrove/beach/ swamp/peat | 481 | 497 | 1175 | 960 | 1000 |
| Urban forest | 240 | 215 | 452 | - | - |
| Agroforestry | 7624 | 13,416 | 15,875 | - | - |
| Non-forest area | 181,594 | 177,151 | 164,006 | 162,500 | 188,168 |
| Total | 200,447 | 198,346 | 200,990 | 188,630 | 395,168 |

Source: MoEF [6].

Figure 2 shows mangrove rehabilitation in Lubuk Kertang, North Sumatra. The mangrove rehabilitation was one of the successful examples of forest restoration in Indonesia. The carbon stored in mangrove biomass increased to 314 tons per ha, and the community income increased from IDR 50,000–100,000/person/day to IDR 100,000–200,000/person/day [96].

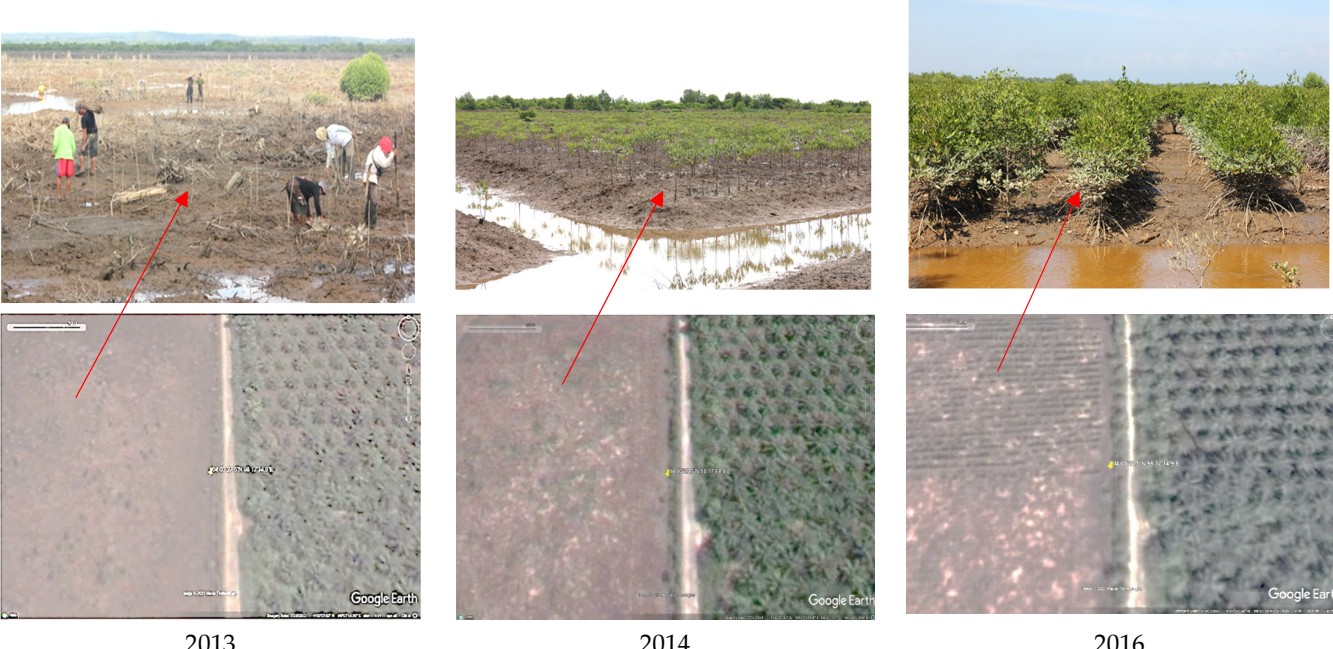

2013          2014          2016

**Figure 2.** Mangrove rehabilitation in Lubuk Kertang, North Sumatra (Source: Arifin [96]).

*4.2. Criteria and Indicators for Forest Restoration*

The concept of forest landscape restoration (FLR) focuses on the impact of restoration efforts, accommodating the integrity of both ecosystem ecology and community livelihoods in the landscape. FLR requires stakeholder collaboration for achieving the restoration target. However, the final goal is not necessarily the return of the original ecosystem but rather increasing the resilience of the forest ecosystem such that it can deliver the function of the forest as a provider of goods and services for the welfare of the community [97].

A comprehensive restoration planning process is needed to ensure success. There must be a clear goal for the restoration and targets for each step in the process [98]. Indicators are important in assessing restoration efforts so that uncertainties and challenges can be managed [21]. To determine the final goals and targets of the FLR, a site reference is needed as a basis for restoration activities and guidance on adaptive management. Various

techniques can be applied to determine site references, such as a spatial approach to look at land cover changes or land degradation [99–102], updating land conditions based on the development of knowledge and land terminology [103], and looking at the degraded ecological components and how to restore them [104]. Some examples of activities carried out based on site references include the use of local plant species in restoration [105], the potential for natural regeneration to accelerate ecosystem recovery [106], and consideration of the welfare of the surrounding community and potential community livelihoods in FLR activities [107,108].

At the indicator level, the criteria used are generally inexpensive, easy to measure, and quick to assess. For example, at the site level, the restoration of forest ecosystems in various forest types generally uses vegetation diversity as an indicator of restoration success compared to ecological processes or reproductive populations, which require commitment, time, cost, and special expertise. Vegetation diversity is relatively easy to inventory, and vegetation is also associated with many functions of forest ecosystems [98]. At the landscape level, biodiversity is generally used as an indicator of restoration success. The list of indicators commonly used to assess the progress of landscape-based restoration activities are presented in Table 5

**Table 5.** List of indicators commonly used to assess the progress of landscape-based restoration activities (compiled from Dey and Schweitzer [98]; other literature cited in this manuscript and from the experience of the authors).

| No. | Component | Indicators |
|---|---|---|
| 1. | Structure | (1) Soils, geology, water bodies; (2) Landform, topography; (3) Habitat connectivity; (4) Community types; (5) Trophic levels; (6) Age, size structure; (7) Peat maturity; (8) Artificial canals; (9) Salinity levels; (10) Peat substrate quality; (11) Historical fire events and drainage; (12) Dissolved organic carbon and dissolved organic nitrogen |
| 2. | Composition | (1) Area of forest and other cover types; (2) Ownership; (3) Biodiversity (e.g., vegetation diversity); (4) Threatened and endangered species; (5) Management unit (KHG, DAS [1]); (6) Pioneer species; (7) Climax species; (8) Root-mats [2]; (9) Cultivation areas; (10) Protective areas; (11) Dispersal agents; (12) Revegetation activity; (13) Soil and ecological legacy |
| 3. | Function | (1) Production of timber, water, wildlife, air; (2) Ecological integrity, health; (3) Economic development; (4) Demographics; (5) Gene flow patterns and migration; (6) Disturbance regime; (7) Nutrient cycling, energy flow; (8) Social well-being and ecosystem services, sustainable livelihood; (9) Viable populations; (10) Policy and law; (11) Recreation; (12) Fragmentation (aggregation index, edge density); (13) Carbon sequestration and storage; (14) Barriers for seawater intrusion; (15) Hydrological integrity, water storage |

[1] KHG = Kesatuan Hidrologi Gambut/Peat Hydrological Unit; DAS = Daerah Aliran Sungai/Watershed; [2] rootmats = term used by the authors to describe the interconnection between compatible roots within the peat ecosystem, forming microtopographical effects and influencing energy flow between species.

### 4.3. Forest and Land Restoration for Soil Conservation and Hydrological Function

Soil erosion is a problem often associated with forest and land degradation [109,110], resulting in decreased soil nutrient cycling and hydrological function [111]. The transformation of natural ecosystems often results in increased erosion [112–114] due to the exposure of the land surface and decreased surface cover. Several soil conservation technologies have been applied to reduce and mitigate soil erosion and surface runoff [115], including restoration activities through several mechanisms such as reforestation, afforestation, agroforestry, etc. Revegetation can help control soil erosion by protecting the soil from direct raindrops and reducing its impact, reducing surface runoff velocity, improving water infiltration, and holding soil particles together [116–118], which has been shown to reduce erosion and runoff surface area by up to 64% and 72%, respectively [119]. Therefore, the management of soil and vegetation is important for controlling the extent of soil loss.

As well as helping to control erosion, FLR in degraded landscapes is expected to improve hydrological function [120] because forests play a very important role in hydro-

logical processes such as regulating fluxes of atmospheric moisture and rainfall patterns over land [121]. Reforestation can sometimes reduce water yields [122], specifically when conducted on pastureland with shallow-rooted annual species, due to the increase in evapotranspiration by deep-rooted forest species [123]. The reduction in water yields is also influenced by the position and extent of the area reforested but is unlikely to be an issue in many areas of the wet tropics, including much of Indonesia, where precipitation exceeds potential evapotranspiration for most months of the year [124]. Where reforestation reduces runoff [125], it can help reduce flooding, depending on the types and age of vegetation chosen [126] and the intensity of the rainfall [116]. Reforestation by enrichment planting was able to recover the hydrological condition of the Tabunio watershed, South Kalimantan [127], as well as in Central Java, Indonesia, after the eruption of Mt. Merapi [128].

Restoration can also help reverse the increased surface runoff and decreased base flow that occurs with clearing [129]. Both of these can be used as indicators to evaluate the success of landscape restoration from a hydrological perspective [130]. Suryatmojo, et al. [131], in Bukit Baka, Central Kalimantan, showed that canopy cover reduction decreases evapotranspiration and increases runoff. Based on many research results, some lessons have been learned about the implementation of forest and land restoration for soil conservation and hydrological function. Even though forests do not always reduce floods, they can slow the speed of water movement through the landscape, which can reduce the peak flow.

*4.4. Forest and Land Restoration for Conservation of Biodiversity*

Biodiversity in Indonesia is managed in nature reserves and nature conservation areas, including 560 conservation area management units (27 Mha) consisting of 212 nature reserves (4 Mha), 80 wildlife reserves (5 Mha), 54 national parks (16 Mha), 133 nature tourism parks (0.8 Mha), 36 forest parks (0.4 Mha), 11 hunting parks (0.2 Mha), and 34 smaller nature conservation areas [132]. While the extent of degraded land in these conservation areas has not yet been fully quantified, around 1.8 Mha of the conservation area is in the form of bare land [133].

Planting to increase tree cover does not necessarily result in the return of ecological functions such as biodiversity, so the area must continue to be monitored to determine forest restoration success. Aerts and Honnay [134] stated that ecosystem restoration must consider above- and belowground biodiversity as an ecosystem unit, known as the biodiversity-ecosystem functioning approach (BEF). However, it is extraordinarily challenging to conduct full ecosystem restoration back to its previous state in terms of biodiversity [135]. Planting using one type of plant may meet the criteria for successful forest restoration required by law but may not be sufficient to spur the return of all biodiversity, especially fauna. Planting with only one tree species has been shown to result in a lower diversity of arthropods, avifauna, amphibians, and small mammals in peat swamp forests, mangroves, and ex-mining areas [118,136,137]. Restoration of mangrove forests in East Java and former coal mines in South Kalimantan with mixed plant species have also been shown to increase the abundance of bird species because birds prefer habitats with higher plant diversity [138,139].

Ecosystem restoration in conservation areas using native plants through enrichment of lowland tropical forest vegetation has been carried out in Manupeu Tanah Daru, Laiwangi Wanggameti [140], and Kutai National Parks [141], while vegetation enrichment in highland tropical forests was carried out in Mt. Ciremai [142,143] and Mt. Pangrango National Parks [144]. Some of the keys to the success of ecosystem restoration using native plants include choosing suitable species according to habitat conditions, performing routine plant maintenance, monitoring growth, scheduling routine patrols, and making firebreaks to prevent fires and illegal grazing of livestock by area managers. In addition, plant invasions will change the uniqueness of ecosystems such as in the savanna of Baluran National Park [145,146], the lowland forest of Ujung Kulon National Park [147,148], and the lowland rain forest of Bukit Barisan Selatan National Park [149]. Several national parks are also

working to restore the carrying capacity of endangered animal habitats for key species such as orangutans, Javan rhinoceros, Sumatran tigers, Sumatran elephants, Sumatran rhinos, and Javan bulls. The restoration of animal habitats is conducted through the enrichment of fodder crops, nesting plants and plants that help arboreal movement, and the habitat control of invasive plant species that interfere with the movement of terrestrial animals (*Acacia nilotica*, *Merremia peltata*) and disrupt animal feed populations (*Arenga obtisufolia*, *Melastoma malabtricum*) [148–153].

Rehabilitation activities are also implemented in mangrove forests. In recent decades, the high dependence and diversity of community use of mangrove forests have led to a decline in biodiversity [154]. A study undertaken on the main islands in Indonesia showed a significant decline in mangrove tree species biodiversity in the mangrove ecosystem [154]: 28 species were recorded on Java in 1993, but only 10 in 2006. Similarly, over the same time on Papua Island around 16 species were lost, and around 11 species were lost on Kalimantan Island [154]. Restoration is needed to restore mangrove ecosystems, and community involvement is a major factor in reducing threats to mangroves [155].

### 4.5. Restoration for Soil Microbial Communities

Soil microbial communities are key to linking above- and belowground ecosystems through nutrient mineralization processes and cycling. Soil microbial diversity is an essential driver of above- and belowground diversity [156]. Soil microbial communities, including bacteria, fungi, arbuscular mycorrhizal fungi (AMF), and actinomycetes, play a significant role in maintaining various ecosystem functions and are considered key indicators of soil health and fertility [157]. Their abundance was found to significantly increase in the topsoil after vegetation restoration [158]. Moreover, soil microbes can reflect the ecological processes that have occurred, so that healthy and resilient soil can help in the formation, growth, productivity, and succession trajectories of local plant communities during restoration. Populations of arbuscular mycorrhizal fungi and soil fungi tend to grow with increasing revegetation and the recovery rate in mined forest and burned peat swamp forest [159,160]. On the other hand, low populations of arbuscular mycorrhizal fungi have been proposed as one of the obstacles to natural regeneration in peat swamp forest restoration [161]. Therefore, changes in and the presence of soil microbial communities need to be considered in forest restoration efforts.

### 4.6. Tropical Landscape Restoration for Climate Change Mitigation in Indonesia

Indonesia holds an important position in the global climate agenda concerning forest-based climate change mitigation. Indonesia has a large area of tropical forests, with globally significant carbon stocks [162,163]. Many studies have been conducted to estimate the carbon stocks of Indonesia's tropical forests. For example, Laumonier, et al. [164] estimated that the aboveground biomass (AGB) of the hill dipterocarp forests of Sumatra ranged from 271 to 478 t/ha or 135 to 239 tC/ha, assuming 50% of dry matter of AGB is carbon, with a mean of 361 t/ha (180 tC/ha). Krisnawati, Adinugroho, Imanuddin and Hutabarat [41] reported that the AGB of primary and secondary dryland forests in Central Kalimantan were 298.2 and 218.2 t/ha, respectively. Tropical lowland forests in Ulu Gadut, West Sumatra stored 482.75 t/ha in AGB [165].

While Indonesia's forests hold globally significant carbon stocks, they are under great pressure due to economic development and population growth. For several decades, large areas of peat swamp forests have been converted to oil palm plantations, releasing around $640 \pm 114$ tCO$_2$e/ha [166]. Butarbutar, et al. [167] found that under conventional logging practices mixed dipterocarp lowland forest emitted 51.18 tC/ha/year. Fire is also a significant threat to Indonesia's carbon stocks, with as much as 0.89 GtCO$_2$e released during the 2015 fire event that affected more than 4.6 Mha across Sumatra, Kalimantan, and West Papua [168]. In South Sumatra, mangrove conversion to coconut plantations resulted in the release of 14.14 MtCO$_2$e to the atmosphere [169]. In addition to deforestation, large areas of tropical forests in Indonesia are under various levels of degradation, contributing

to the increase in GHGs [170,171]. Considering the large coverage of tropical forests with high carbon content and the ongoing forest disturbance with significant carbon emission consequences, it is important to conserve Indonesia's remaining tropical forests and restore those that are degraded to support climate change mitigation efforts.

Recent studies have suggested that achieving the Paris Agreement's goals on climate change mitigation will require a significant contribution from Indonesia's forests [172,173]. Currently, the main driver of FLR adoption is the need to restore biodiversity and mitigate climate change [174]. As a means of climate change mitigation, FLR should be integrated with development program planning—it needs to be part of a program of sustainable development that is integrated into national policies [175]. Similarly, Tacconi and Muttaqin [176] argued that forest rehabilitation, as part of a program to reduce emissions from forests, requires the implementation of policies and actions at the national level. One of the efforts is the establishment of the national action plan for greenhouse gas emission reduction (*Rencana Aksi Nasional Penurunan Emisi Gas Rumah Kaca/RAN-GRK*), coordinated by the MoNDP, which is the starting point for the central government, regional governments, communities, and economic actors to reduce GHGs to under national development targets. In response to the Paris Agreement on climate change, the government of Indonesia renewed its GHG emission reduction target through Indonesia's nationally determined contribution (NDC), to 29% unconditional and 41% conditional, with projected business as usual (BAU) emissions of 2869 $MtCO_2e$ in 2030.

The achievement of GHG emission reductions reported by MoNDP [177] in 2017 was 24%, of which 21% came from the contribution of forestry and peatland. The GHG emission in BAU baseline was 1860 $MtCO_2e$ in 2017, with cumulative emissions during 2010–2017 of 13,030 $MtCO_2e$. Net GHG emission potency in 2017 was 1465 $MtCO_2e$ after emission reduction from the forestry and peatland sector, and 1408 $MtCO_2e$ with an emission reduction of all sectors (Figure 3). A recent report by MoEF [6] stated that the GHG emission reduction of the forestry sector in 2018 was only 37 $MtCO_2e$ against the baseline. This figure was far less than in 2017 due to an escalation in forest and peatland fires in 2018.

The contribution of the forestry sector to Indonesia's NDC was based on avoiding deforestation and forest degradation, as well as reforestation. The GHG emission reduction from avoided deforestation and forest degradation reported during 2016–2017 was 8598 Gt $CO_2e$ and 8680 Gt $CO_2e$, respectively [6]. On the reforestation side, the MoEF has an annual target of 800,000 ha with a 90% survival rate. A total of 1.18 Mha of tree plantings were reported during 2015–2019, including conservation/protected forest, mangrove/swamp/peat area, urban forest, agroforestry, and land rehabilitation with seedlings from community nurseries. This achievement was lower than the target due to a limited budget allocation that was only sufficient to establish around 200,000 ha/year [6].

Reviewing the recent state of GHG emission reduction against the unconditional 29% target for 2030 suggests that the target is already ambitious. In particular, FLR activities directed at supporting climate change mitigation need to consider several factors. Previous studies reveal that the implementation of FLR in Indonesia in the context of climate change mitigation requires a clear link between action and intended outcome [178], considering local perspective and impact [175,179,180], introducing conservation practices [181,182], and involving small growers [183].

Small-scale forest plantations can contribute to forest and land restoration in the context of climate change mitigation as well as rural economic enhancement [183], so it is important that small growers be engaged in the FLR action. However, GHG emission reduction-based plantation programs also require conservation awareness beyond socioeconomic need. Ignoring such awareness would result in substantially lower GHG emission reduction, particularly from a land-use change and forestry (LUCF) perspective. People will continue to cultivate their land as usual without considering long-term ecosystem restoration. Therefore, Malahayati and Masui [181] suggest that introducing conservation practices is needed to achieve a successful tree plantation program that also contributes to climate change mitigation.

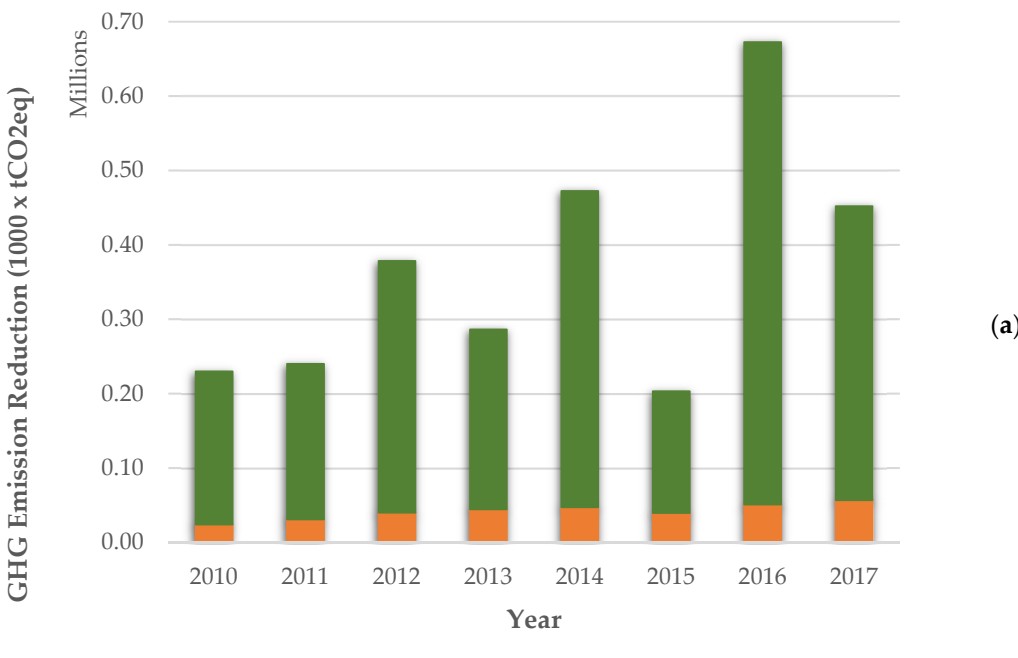

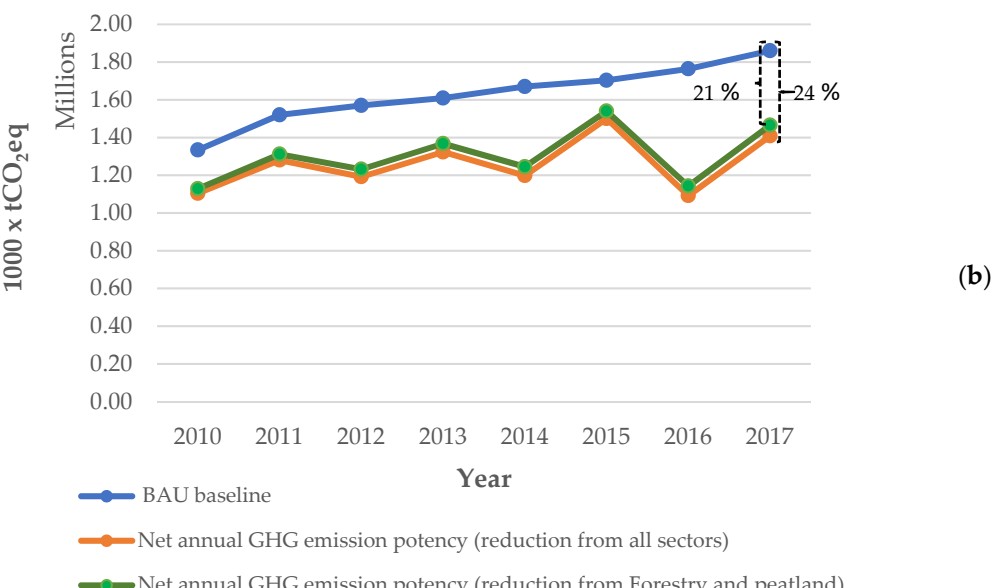

**Figure 3.** Achievement of GHG emission reduction by sector (**a**) and scenario (**b**); Source: adapted from MoNDP [177].

### 4.7. Livelihood Benefits from Forest Restoration

Although forest landscape restoration is a relatively new concept in Indonesia [10,78], forest restoration initiatives have been conducted in Indonesia for a long time through various programs connected with forest ecosystem improvement and social benefits from the forests. FLR requires greater stakeholder involvement to ensure that reforestation programs can provide both short- and long-term benefits for the ecosystem and community livelihoods [10]. Forest landscape restoration places the community at the center of the process. It combines multiple approaches to the social-ecological system, emphasizing the expected improvement in local livelihoods, well-being, and resilience for the commu-

nity [16,184,185]. Beyond the biophysical condition and technical aspects, improvement and sustainable livelihoods of the community are critical for forest landscape restoration. Livelihood benefits derived from forest restoration are contextual and can potentially be generated from a range of areas, including reforestation and afforestation, nontimber forest products including forest ecosystem services, access, and land rights of community forestry [186,187].

Permits to access and use the forest through community forestry initiatives can encourage the community to take an active role in forest restoration for livelihood improvement. Community forestry in Indonesia has had a long history of forest management for both social and ecological outcomes since the 1970s [188–190]. Evolving community forestry programs in Indonesia have improved the legal access of the community to forest management to fulfill their livelihood needs as well as maintain and improve the forest cover [191–195]. Communities play an increasing role in forest restoration too, with the granting of access to the forest being a pathway to involve the community in FLR as well as meeting their livelihood needs. However, some studies note that community forestry has not been universally successful, with some schemes seeing low levels of community participation, low-capacity development, a lack of technical support, low levels of funding, and an unclear access mechanism [196–200]; these issues need to be explored and understood in order for community forestry to be able to best support FLR.

In the context of FLR and rural livelihoods, agroforestry has the capacity to contribute to both social and ecological outcomes [201,202]. While improving tree cover in the landscape, agroforestry can also support enhancing and diversifying community incomes, improving food security, providing environmental services, and improving resilience in hazard–prone areas [20,202–210]. In addition, diverse commodities and services that can improve livelihoods can also be acquired from nontimber forest products (NTFPs) and forest ecosystem services in the FLR framework. NTFPs and forest ecosystem services have contributed to the community's livelihoods in some studies [20,57,138,180,202,211–214]. Local communities are the main beneficiaries of NTFPs and forest ecosystem services, and developing those commodities is connected tightly with the community livelihoods surrounding the forest in the context of FLR implementation.

Livelihood improvement from peatland restoration received greater attention after the Indonesian government established the Peatland Restoration Agency (PRA) in early 2016. Through the revitalization program, PRA and its partners launched a number of programs to improve the livelihoods of people on peatlands, including land-based, water- and fishery-based, and environmental services programs [73,215]. The large scale of the peatland restoration project requires a significant number of stakeholders to be involved [123,216], including government agencies, industrial corporations, small and large landholders, NGOs (nongovernmental organizations), and scientists [216,217]. The implementation of the restoration action policy must also be more comprehensive, such that the rewetting, revegetation, and revitalization (3R) programs are carried out at the same location so the results and impacts are more significant for the community and environment. Rewetting and revegetation activities may help promote biodiversity and facilitate other objectives, including ecosystem services and socioeconomic benefits for community livelihood. This would allow multiple outcomes to be realized simultaneously [216]. If there is competition for land to achieve either conservation or development goals, one or the other is going to suffer, or a suboptimal outcome will be achieved [174].

Forest landscape restoration (FLR) improves livelihoods for the community through their direct involvement and the reestablishment of fully functioning ecosystems [52]. Including indigenous communities and local stakeholders in the FLR process is necessary for achieving ecological restoration and improving human well-being [186,216]. The community will be willing to cooperate or be actively involved in a program if they understand the benefits, either direct or indirect [218,219]. In the case of Kayu Labu village, South Sumatra, for example, engagement of local people in several government programs related to restoration remains low because information about the implementation and benefits

of these programs has not been strongly conveyed to the wider community [220]. In fact, peatland restoration activities will substantially benefit the community because they can reduce the impact level of fires in the dry season and flooding in the rainy season [73], as well as provide a source of nontimber forest products [221].

The livelihoods project in Indonesia has also demonstrated that the restoration of degraded wetlands can go hand-in-hand with improving family income and reducing poverty. Mangrove restoration projects that are spread across many coastal locations such as Aceh, North Sumatra, East Java, East Kalimantan, South Sulawesi, and others [222–224] have generated new sources of income and restored mangrove forests that could increase the safety net of the local population [225]. Local villagers were able to increase their income by selling products from the restored mangrove areas such as giant tiger prawns (*Penaeus monodon* Fabricius), soft-shell crabs, fish, seaweed, mollusks, batik dye, and honey [226,227]. Restoring mangrove plantations in partnership with local NGOs and other stakeholders has improved coastal protection and revived biodiversity [228]. Forest landscape restoration can safeguard biodiversity by taking a landscape approach using appropriate technologies with practical applications and producing real benefits for communities by working in partnership with related stakeholders [229,230].

FLR, as a long-term and complex process, should involve the local community as the main actor. Engagement with the local community is one of the important factors in connecting FLR with the improvement of livelihoods at the field level. Local communities are key to the success of forest restoration and the stakeholders with the most to gain from the projects [216]. Active participation of the local community is a promising approach for improving livelihoods in the context of FLR [20,231]. In the context of a dynamic landscape, building a network of stakeholders is essential to obtain the ecological and livelihoods goals of FLR. Restoring a landscape requires planning on a large scale, long time frames, and continuously meeting the needs of local people and other stakeholders through the restoration process [232–235]. Therefore, livelihood revitalization through strengthening the institution, community empowerment, stakeholder participation, and market certainty will greatly improve the success of forest landscape restoration [236]. With more stakeholders engaged in the forest landscape, harmonization and synchronization of interests are important for realizing the benefits of restoration activities. Multistakeholder networks and collaborative action programs are valuable options to generate livelihood benefits from FLR.

### 4.8. Economics of Forest Restoration

Forest landscape restoration (FLR) is generally costly and faces several challenges since it requires a complex range of inputs covering ecological, technical, social, and economic aspects [52,237,238]. Restoration typically involves a large investment with a high risk of failure and a long lead time to achieving the benefits [237]. Therefore, many parties are reluctant to invest in restoration unless they are obligated to do so [239]. In Indonesia, restoration projects are largely dependent on international donors and the private sector for funding [240], but the government also plays a critical role in setting the policy and provision of the budget.

Integrating ecology and economics in ecosystem restoration is indispensable to ensure cost-effective programs [60,241]. Payment for ecosystem services is a market-based system that can promote restoration and save costs [60] because it focuses on restoring those services that are important to communities [242]. Several studies across Indonesia have found a net positive economic benefit of restoration, including Yanarita, et al. [243], who assessed the net present value (NPV) of the agroforestry system of *Metroxylon sagu* in Central Kalimantan and found that it was financially feasible as a peatland restoration option. Communities whose livelihoods depend on the natural productivity of mangrove forests in Balikpapan Bay, East Kalimantan, have gained economic benefits from restoration activities through increased harvesting of wood and fish products [244]. Forest restoration in Sumatra has been shown to increase income from natural forest resources for indigenous

people nearby [213,245], and restoring the protected forest ecosystem of Sungai Bram Itam protected peatland forest in Tanjung Jabung Barat Regency, Jambi Province, with the agroforestry system of *Shorea balangeran*, *Litsea* spp., *Syzygium* sp., *Eugenia* spp., *Vitex* sp., *Illex cymosa*, *Dyera polyphylla*, and *Areca catechu*, resulted in increased economic returns to the community [246].

The cost-benefit analysis (CBA) framework is generally applied in the economic analysis of landscape restoration, including agroforestry, afforestation, reforestation, and assisted natural regeneration practices [52]. Another framework is a cost-effectiveness analysis (CEA), which compares projects that have similar goals. However, it is difficult to compare different levels of restoration, and costs and benefits that occur in the projects are difficult to quantify [247].

Restoration finance standards remain inadequate due to the varying contexts, interventions, and objectives of restoration projects, as well as the absence of clear standards and protocols for the collection of cost and benefit data [248]. The economics of ecosystem restoration (TEER) is a framework that was developed to assess the costs and benefits of specific restoration projects and interventions, which was then broadened to more generally examine the economics of ecosystem restoration [248]. It has been used to help ensure that the appropriate cost and benefit data required for an analysis are collected at all stages in the restoration process to better quantify the scale of restoration funding needs and to inform decisions on the allocation of resources [248,249]. The flow of benefits from the restored ecosystem may provide a positive return on investment. However, Rahmawati [250] found that the benefits of natural ecosystems are often poorly recognized, and hence it is difficult to attract the level of funds or investment needed for ecosystem restoration.

Several conditions are needed to facilitate the success of FLR, including secure funding for activities over multiple years, minimal bureaucracy, constructive regulations [251], a positive mindset from local proponents [252], flexibility to adapt to the planting season and local conditions, economic feasibility, participation of the surrounding community, transparent and measurable progress with well-planned stages, and steps to ensure the sustainability of the outcomes [253].

*4.9. Monitoring and Evaluation of the Implementation of Forest Restoration*

An effective monitoring and evaluation (ME) program contributes to the success of forest restoration by measuring progress, determining corrective actions, and modifying activities that are needed to achieve the long-term goals.

Mansourian, et al. [254] promote an ME framework that can be used to monitor the success of forest restoration based on several aspects of the system, namely: (1) nature (diversity and ecosystem function), (2) environmental benefits, and (3) livelihoods and well-being. Moreover, to evaluate the success of forest restoration, Ruiz-Jaen and Aide [255] suggest using at least two variables in each of three attributes of ecosystem function obtained from comparing two reference conditions. They suggest three ecosystem attributes that should be assessed in monitoring the success of forest restoration: (1) species diversity (plants, arthropods, amphibians, reptiles, birds, etc.); (2) vegetation structure (cover, density, biomass, and height); and (3) ecological processes (nutrient pool, organic matter, and presence of mycorrhizae) [255].

Through the Ministry of Forestry, the government of Indonesia (GoI)) stated that restoration activity success can be assessed based on the presence of species, vegetation structure, and population dynamics that resemble the reference ecosystem or original conditions. Six characteristics of a target ecosystem that has been successfully restored are presented in Figure 4.

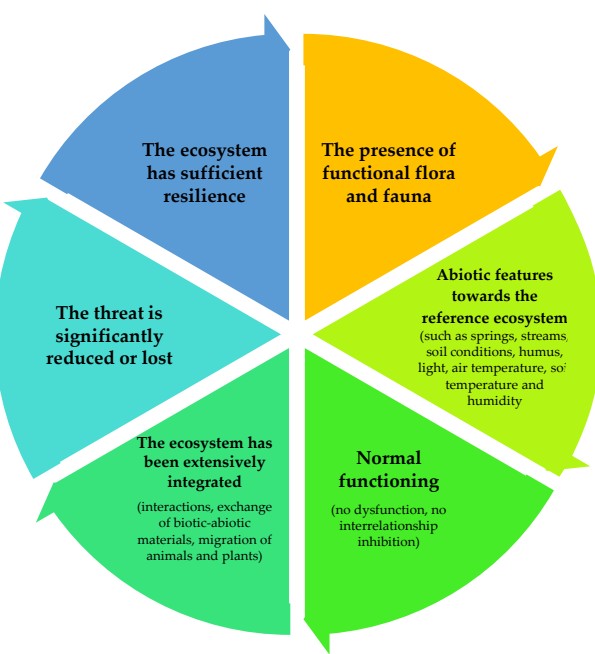

**Figure 4.** Characteristics of an ecosystem that has been successfully restored; (Source: Ministry of Forestry Regulation No. 48 of 2014).

Achieving the goals of forest restoration in Indonesia requires not only the restoration of ecosystems related to biodiversity and ecological processes but also the improvement and sustainability of livelihoods, considering that forest encroachment by local communities has occurred in various areas and human coexistence with the forest needs to be promoted. Precisely how restoration efforts can be accompanied by the fulfillment of livelihood needs is a key question. Moreover, selecting multiple functional species that can support ecosystem recovery and support food security and the livelihoods of local communities is crucial in FLR. Another strategy would be encouraging business opportunities through ecotourism activities connected with restoration projects. Successful ecosystem restoration requires support for sustainable livelihood options.

## 5. Silvicultural Techniques for Tropical Forest Restoration

FLR involves not only tree planting but also reestablishing the overstorey as an important first step. Holding to that principle, this section focuses on the silvicultural techniques that have been implemented in tropical forest restoration in Indonesia, including six primary forest ecosystems: lowland and upland tropical rain forest, monsoon forest, savanna, peat swamp forest, and mangrove forest. The lowland tropical rain forests in Indonesia are mainly dominated by dipterocarp species and have some of the world's highest terrestrial ecosystem biodiversity [256,257]. The upland forest is similar to a tropical rain forest but with less diversity of tree species (mostly Lauraceae and Fragaceae) and the subalpine rain forest has common tree genera of *Araucaria*, *Dacrydium*, *Podocarpus*, and *Quercus* [258].

Tropical monsoon or seasonal forests experience a 3–4-month dry season [259], mostly dominated by acacia and albizia and characterized by lower tree diversity, lower canopy height, fewer canopy strata, smaller leaf area index, lower plant biomass, lower net productivity, and lower tree diameter growth rate [260].

Savanna is a mixed woodland-grassland ecosystem typically characterized by a continuous cover of $C_4$ grass and sparse cover with woody plants but no closed canopy [261]. This ecosystem is mostly found in East Nusa Tenggara, as well as small areas in West Nusa Tenggara and Java Island [262]. The floral composition of the savanna is influenced by precipitation, fires, and grazing [263].

The other two main types of forest ecosystem in Indonesia are wetland ecosystems: peat swamp forest and mangrove forest. Peat swamp forest is a forest ecosystem formed

from partially decomposed organic matter (peat) under inundated conditions [264], with 47% of the world's tropical peatland located in Indonesia, primarily in Kalimantan and Sumatra [265]. The last major type of forest, mangroves, are a group of trees and shrubs that grow in intertidal environments along tropical coasts [266]. Approximately 22.6% of the world's mangrove ecosystem is located in Indonesia [267]. Although these forest ecosystems are different from each other, the silvicultural techniques for restoring such ecosystems follow general rules, including passive restoration, active restoration, and passive and active restoration [217,268,269].

Passive restoration relies on natural regeneration. To be successful, there needs to be a ready supply of propagules (sprouting from adjacent trees) or seeds, either dispersed from adjacent sources or from the soil seed bank [106,217,268,269]. Natural regeneration has significant potential for the restoration of large areas at a low cost. However, it will only occur where there is limited or no disturbance during the restoration process, and the ecosystem has not crossed an ecological threshold and moved to a new state condition [123]. Moreover, Lamb [123] emphasized that passive restoration is unsuitable for sites that are too extensively degraded, have too low a tree density or no remnant adjacent forest patches, or have too few seed dispersers.

Active restoration involves the planting of seeds or seedlings of desirable species [269–271]. A combination of active and passive restoration or assisted natural regeneration involves clearing weeds to reduce competition with desired seedlings or sprouts [272] or conducting enrichment planting to maintain the diversity of late-successional species [273]. There are four main factors that determine the success of active restoration: (1) production of good-quality planting stock [126,268,269,274]; (2) site-species matching [268,273–275]; (3) postplanting site maintenance [268,274]; and (4) level of site degradation [73,107,276].

Due to the severely degraded condition of most tropical forest areas in Indonesia, restoration efforts are mostly active through planting or enrichment planting. The success of forest restoration is also determined by key silvicultural factors such as the availability of planting stock, preplanting activities, planting activities, and maintenance. In the upland forest, vegetative propagation, such as from shoot cuttings, is typically the method of choice for recalcitrant seeds such as *Araucaria* [277], while generative propagation is chosen for orthodox seeds such as pine (*Pinus merkusii*) [278]. Moreover, in upland forests, a monoculture planting pattern is often chosen, especially for species such as pine, with higher production values for both NTFPs and timber [278]. However, pine is also intercropped with agricultural plant species when the location of forest restoration is adjacent to forest-dwelling communities [278,279]. Due to steep slopes, restoration of upland forests generally requires terrace application [280] or conservation tillage [281] to prevent soil erosion.

In monsoon forests, the choice of planting stock can be more varied; some species can be collected as wildlings (*Toona* sp., *Maesopsis eminii*, mahogany) because those are adaptive species, and the seeds are easy to disperse [282,283]. Clonal seed orchard seed [284] and coppice [283,285,286] have been available for teak (*Tectona grandis*). For *Albizia chinensis*, seedlings grown from a Papuan provenance (Wamena, Biak, Indonesia) are typically chosen due to their good growth performance [287]. These species are planted in two patterns: monoculture and intercropping. Monocultures are planted when the location is far from settlements, and in rocky and steep landscapes with infertile soil [288]. Intercropping is typically chosen in locations adjacent to settlements, which often have better soil fertility, and in flatter parts of the landscape [288]. Postplanting activities for monsoon forest restoration include fertilization, especially for *Albizia* sp. [289], host plant planting for *Santalum album* [290], and mulching to suppress weed growth and maintain soil moisture [291].

Restoration of savanna ecosystems was conducted in Baluran East Java and East Sumba using seedlings as planting stock for local grasses, local shrubs, and shelter trees [123,292]. Pre and postplanting weed control was required to suppress *Acacia nelotica* weed growth in Baluran savanna, using a combination of slash-and-burn and chemical control [292–296]. Controlled burning of the East Sumba savannah was carried out using firebreaks and installing bioreferences/living fences at the boundaries [123,297–299].

Lowland tropical rain forest restoration has employed varied planting stock, including seedlings [273], wildlings [274,300], and shoot cuttings using the KOFFCO (Komatsu-FORDA Fog Cooling System) method for mass propagation of shoot cuttings [301]. Gap planting and line planting are two establishment options in lowland tropical rain forest restoration [302]. Gap planting is preferable for light-demanding species such as *Shorea acuminata* [302]. Postplanting activities include weeding and thinning, especially liberation thinning of early successional species and liana cutting to enhance the light penetration for optimum growth [274,303,304].

In peat swamp forest, the key preplanting activity has been rewetting the area through the construction of infrastructure such as canal blocks, and canal backfilling in Riau, Jambi, South Sumatra, and West, Central, and South Kalimantan [73,107,276,305]. Appropriate hydrological construction allows the water table to increase [276,305]. Seedlings, wildlings, and cuttings are the key planting stock that has been used in tropical peat swamp forest restoration [107,306]. Seedlings raised from seeds, rather than wildlings, have higher success rates, provided that an appropriate germination technique is employed, and a suitable mycorrhizal inoculant is available [73,108,306]. Planting into mounds and raised beds can help with survival under inundation conditions. Another important activity is to increase the bulk density of the peat around the seedling by chopping the peat and compacting it around the newly planted seedlings [73,307–309]. Weed control three times in the first year after *Shorea balangeran* planting increased height growth by 40% and diameter growth by 50% [308]. Mycorrhizal fungi improve the capacity of the plants to find sources of nutrients in peatlands, both in flooded and dry conditions, with good growth responses, both in the nursery and in the field [309,310].

In mangrove forest restoration, propagules can be directly sown. However, planting seedlings or wildlings is more common in mangrove restoration to ensure a higher survival rate [230,311,312]. There are a range of planting techniques for mangrove restoration, including the *cemplongan* (circle land clearing) method, bamboo poles, water breaks, raised beds, huge polybags, muddy ditches, clusters, huge muddy halls, and agroforestry [312]. Of these, the cemplongan method had the best results in terms of plant growth when planted at $0.5 \times 0.5$ m, while agroforestry techniques such as agrosilvofishery and agrosilvopasture had the best results for degraded mangroves surrounded by landless poor communities [311,312]. Weeding and fertilization with combinations of rock phosphate, a humic substance complex and foliar nutrient are two postplanting activities to improve the growth of seedlings [312].

A list of local plant species used in tropical forest restoration in Indonesia, including lowland tropical rain forest, upland forest, monsoon forest, savanna, peat swamp forest, and mangrove forest, is presented in Table 6.

**Table 6.** Plant species employed in tropical forest restoration in Indonesia.

| No. | Species | Family | Remarks | Reference |
|---|---|---|---|---|
| | | **Lowland Tropical Rain Forest** | | |
| 1 | *Terminalia catappa* | Combretaceae | Shade plant, fast-growing | |
| 2 | *Enterolobium cyclocarpum* | Fabaceae | Shade plant, fast-growing Bird fodder | [313] |
| 3 | *Samanea saman* | Fabaceae | Shade plant, fast-growing Bird fodder | |
| 4 | *Shorea balangeran* | Dipterocarpaceae | Local species Protected species | |
| 5 | *Melaleuca leucadendron* | Myrtaceae | Local plant, fast-growing Bird fodder | |
| 6 | *Dyera lowii* | Apocynaceae | Local species | |
| 7 8 9 | *Shorea macrophylla* *S. parvifolia* *S. seminis* | Dipterocarpaceae | Sensitive to light intensity; significant height and diameter growth in first year after planting | [314] |

**Table 6.** *Cont.*

| No. | Species | Family | Remarks | Reference |
|---|---|---|---|---|
| 10 | *Dryobalanops beccarii* | | High light intensity is able to boost | |
| 11 | *Parashorea macrophylla* | Dipterocarpaceae | height and diameter growth up to | |
| 12 | *S. macrophylla* | | 24–81 months after planting | |
| 13 | *Scorodocarpus borneensis* | Olacaceae | Timber-producing species | |
| 14 | *Quercus* sp. | Apocynaceae | | |
| 15 | *Dipterocarpus* sp. | Dipterocarpaceae | | |
| 16 | *Shorea leprosula* | Dipterocarpaceae | | |
| 17 | *Dryobalanops oblongifolia* | Dipterocarpaceae | | |
| 18 | *Mangifera indica* | Anacardiaceae | | |
| 19 | *Artocarpus integra* | Moraceae | Fruit-producing species | |
| 20 | *Ficus* sp. | Moraceae | Deep root, fruit attracts animals | [315] |
| 21 | *Albizia falcataria* | Fabaceae | | |
| 22 | *Anthocephalus chinensis* | Rubiaceae | | |
| 23 | *Duabanga moluccana,* | Hypericaceae | | |
| 24 | *Eucalyptus deglupta* | Myrtaceae | Commonly found in secondary forests, a good reference for rehabilitation | [316] |
| 25 | *Macaranga gigantea* | Euphorbiaceae | | |
| 26 | *Octomeles sumatrana* | Datiscaceae | | |
| 27 | *Peronema canescens* | Lamiaceae | | |
| 28 | *Durio* spp. | Malvaceae | | [315] |
| 29 | *Artocarpus* spp. | Moraceae | Preferred by community | |
| 30 | *Eusideroxylon zwageri* | Lauraceae | | |
| 31 | *Shorea leprosula* | Dipterocarpaceae | | |
| 32 | *Senna siamea* | Fabaceae | | [314] |
| 33 | *Peronema canescens* | Verbenaceae | | |
| 34 | *Shorea macrophylla,* | Dipterocarpaceae | | |
| **Mangrove Forest** | | | | |
| 1 | *Avicennia marina* | Avicenniaceae | | |
| 2 | *Avicennia officinalis* | Avicenniaceae | | |
| 3 | *Bruguiera cylindrica* | Rhizophoraceae | | |
| 4 | *Bruguiera gymnorhiza* | Rhizophoracae | | |
| 5 | *Bruguiera sexangula* | Rhizophoraceae | | [317] |
| 6 | *Ceriops tagal* | Rhizophoraceae | | |
| 7 | *Excoecaria agallocha* | Euphorbiaceae | | |
| 8 | *Heritiera littoralis* | Sterculiaceae | | |
| 9 | *Lumnitzera racemosa* | Combretaceae | | |
| 10 | *Rhizophora apiculata* | Rhizophoraceae | | |
| 11 | *Rhizophora mucronata* | Rhizophoraceae | | |
| 12 | *Rhizophora stylosa* | Rhizophoraceae | | |
| 13 | *Scyphiphora hydrophyllacea* | Rubiaceae | | |
| 14 | *Xylocarpus granatum* | Meliaceae | | |
| **Peat Swamp Forest** | | | | |
| 1 | *Gonystylus bancanus* | Thymeleaceae | | |
| 2 | *Shorea teysmaniana,* | Dipterocarpaceae | Local timber species | |
| 3 | *Shorea pauciflora* | Dipterocarpaceae | | |
| 4 | *Shorea belangeran* | Dipterocarpaceae | | |
| 5 | *Calophyllum macrocarpum* | Callophylaceae | | |
| 6 | *Palaquium* spp. | Sapotaceae | | |
| 7 | *Dacrydium elatum* | Podocarpaceae | | [318] |
| 8 | *Agathis bornensis* | Araucariaceae | | |
| 9 | *Lopopethalum multinervium* | Celastraceae | | |
| 10 | *Tetramerista glabra* | Theaceae | | |
| 11 | *Alstonia pneumatophora* | Apocynaceae | | |
| 12 | *Dyera polyphylla* | Apocynaceae | Local, fast-growing species | |

**Table 6.** *Cont.*

| No. | Species | Family | Remarks | Reference |
|---|---|---|---|---|
| | | **Monsoon Forest** | | |
| 1 | *Aegle marmelos* | Rutaceae | | |
| 2 | *Albizia procera* | Leguminosae | | |
| 3 | *Alchornea rugosa* | Euphorbiaceae | | |
| 4 | *Alstonia scholaris* | Apocynaceae | | |
| 5 | *Alstonia spectabilis* | Apocynaceae | | |
| 6 | *Buchanania arborescens* | Anacardiaceae | | |
| 7 | *Canarium acutifolium* | Burseraceae | | |
| 8 | *Canarium asperum* | Burseraceae | | |
| 9 | *Dillenia pentagyna* | Dilleniaceae | | |
| 10 | *Dysoxylum caulostachyum* | Meliaceae | | |
| 11 | *Ficus hispida* | Moraceae | | |
| 12 | *Ficus racemosa* | Moraceae | | |
| 13 | *Ficus septica* | Moraceae | | |
| 14 | *Ficus variegata* | Moraceae | | |
| 15 | *Garuga floribunda* | Burseraceae | | |
| 16 | *Homalanthus giganteus* | Euphorbiaceae | | |
| 17 | *Homalanthus populneus* | Euphorbiaceae | Adapted to dry conditions | [319] |
| 18 | *Homalium bhamoense* | Euphorbiaceae | | |
| 19 | *Litsea glutinosa* | Lauraceae | | |
| 20 | *Macaranga tanarius* | Euphorbiaceae | | |
| 21 | *Mallotus richinoides* | Euphorbiaceae | | |
| 22 | *Melia azedarach* | Meliaceae | | |
| 23 | *Melochia umbellata* | Malvaceae | | |
| 24 | *Pipturus argenteus* | Urticaceae | | |
| 25 | *Planchonia* sp. | Lecythidaceae | | |
| 26 | *Planchonia valida* | Lecythidaceae | | |
| 27 | *Polyscias aherniana* | Araliaceae | | |
| 28 | *Pterospermum diversifolium* | Sterculiaceae | | |
| 29 | *Rhus taitensis* | Anacardiaceae | | |
| 30 | *Sagaraea lanceolata* | Annonaceae | | |
| 31 | *Schleichera oleosa* | Sapindaceae | | |
| 32 | *Tetrameles nudiflora* | Datiscaceae | | |
| 33 | *Timonius timon* | Rubiaceae | | |
| 34 | *Trophis philippinensis* | Moraceae | | |
| | | **Savanna** | | |
| 1 | *Dichanthium caricosum* | Poaceae | A perennial grazing pasture with excellent ground cover | |
| 2 | *Polytrias amaura* | Poaceae | It issued as a lawn grass in tropical and subtropical regions | |
| 3 | *Heteropogon contortus* | Poaceae | A valuable pasture species | |
| 4 | *Themeda* spp. | Poaceae | Native to Southeast Asia | [292] |
| 5 | *Sclerachne punctata* | Poaceae | Dominant at Baluran Savanna, East Java | |
| 6 | *Brachiaria reptans* | Poaceae | | |
| 7 | *Azadirachta indica* | Meliaceae | Fast-growing species | |
| 8 | *Schleichera oleosa* | Sapindaceae | | |
| 9 | *Acacia leucophloea* | Fabaceae | Leaf for fodder | |
| 10 | *Tamarindus indica* | Fabaceae | Produces edible fruit | |
| | | **Upland tropical rain forest** | | |
| 1 | *Pinus merkusii* | Pinaceae | Sap and wood producers, land and water conservation | [278] |
| 2 | *Agathis loranthifolia* | Araucariaceae | Sap producer | |
| 3 | *Dacrydium* sp. | Podocarpaceae | Local timber species | [320] |
| 4 | *Podocarpus* sp. | Podocarpaceae | Local timber species | |

**Table 6.** *Cont.*

| No. | Species | Family | Remarks | Reference |
|---|---|---|---|---|
| 5 | *Casuarina junghuhniana* | Casuarinaceae | Local timber species | [321] |
| 6 | *Araucaria cuninghamii* | Araucariaceae | Sap and wood producer | [322] |
| 7 | *Toona sureni* | Meliaceae | Wind break | |
| 8 | *Khaya anthoteca* | Meliaceae | Local timber species | [323] |
| 9 | *Coffea arabica* | Rubiaceae | Food | |

## 6. Conclusions

While forest landscape restoration is a relatively new concept in Indonesia, forest rehabilitation practices have a long history. To boost its economy, Indonesia has rapidly exploited its forests on the outer islands since the mid-1960s, and as a result, it has become one of the world leaders in tropical timber exports. However, the resultant deforestation and forest degradation have left a legacy of environmental issues, including habitat degradation and loss of biodiversity, water quality and quantity deterioration, air pollution, and greenhouse gas emissions that make a globally significant contribution to climate change. The government has recognized this issue and enacted various regulations aimed at restoring forest ecosystems, followed by the implementation of many rehabilitation/restoration programs across a wide range of ecosystems. Because most tropical forest areas in Indonesia are severely degraded, restoration efforts have primarily focused on active restoration through planting or enrichment planting, even though this is more expensive than passive restoration. Production of good quality planting stock, level of site degradation, site-species matching, and postplanting site maintenance are factors determining the success of the active restoration. Restoration governance has improved mostly in the wetland ecosystem (i.e., peatland and mangrove ecosystems) through various regulatory instruments that place more emphasis on sustainability than on economic considerations.

FLR recognizes that the key to sustainable restoration is changing fundamental socioeconomic drivers and focusing on key ecosystem services. Understanding and solving socioeconomic and institutional problems will substantially influence the success or failure of a restoration program. Moreover, the active participation of the community is critical to the success of restoration planning and implementation. Hence, in-depth research on socioeconomic aspects of forest landscape restoration in Indonesia is still needed. The concept of FLR holds significant promise for Indonesia to simultaneously improve the restoration of key ecosystem services as well as ensure that community livelihoods are maintained or improved through the process, thus maximizing the chances of real and long-lasting forest restoration. We conclude that successful forest landscape restoration must be based on scientific knowledge, be socioeconomically relevant, and not take place without the consent and engagement of local communities.

**Author Contributions:** Y.I., T.W.Y., S.L., B.W., B.H.N., H.Y.S.H.N., D.R., P., M.T., R.N.A., E.S., P.B.P., P.B.S., N.P.N., S.A.C., R.S.W., R.P., W.H., M.S., A.W. (Ary Widiyanto), M.M.B.U., S., A.W. (Aji Winara), T.W. and D.M., had an equal role as main contributors in discussing the conceptual ideas and the outline, providing critical feedback for each section, and writing the manuscript. All authors have read and agreed to the published version of the manuscript.

**Funding:** This research was partly supported by ACIAR project FST/2016/144.

**Institutional Review Board Statement:** Not applicable.

**Informed Consent Statement:** Not applicable.

**Data Availability Statement:** Not applicable.

**Acknowledgments:** We would like to thank ACIAR FST/2016/144 for providing financial support for this paper. We would like to thank Zainal Arifin, the Director of Soil and Water Conservation, Directorate General of Watershed Management and Forest Rehabilitation, MoEF for providing us with supporting data for this review. We thank three anonymous reviewers for providing helpful comments on earlier drafts of the manuscript.

**Conflicts of Interest:** The authors declare no conflict of interest.

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
