# Peer review of "Tropical Forest Landscape Restoration in Indonesia: A Review"

_land, doi:10.3390/land11030328_

Round 1

Reviewer 1 Report

L149-153 “The aim of this paper was to examine the regulatory, institutional, and policy aspects of forest restoration in Indonesia, as well as the implementation of forest restoration across the country. We explore the failures, success stories, and criteria and indicators that have been developed for forest restoration. We also discuss the latest (best practice) technologies for successful forest landscape restoration.”   These sentences also show up in the abstract.

The purpose of this paper as written in lines 149-153 does not clearly capture the content.  With some minor text adjustments throughout, this would be a comprehensive publication.  

One need is to explain in the introduction the connection between forest landscape restoration, and forest restoration, and how forest landscape conditions in Indonesia necessitates consideration of peatlands, wetlands, savannahs, etc, and mangroves.  It makes sense to discuss forest landscapes which includes non-forest areas in the ecosystems of Indonesia, but this is not made clear.  As written, to read this is about forests and yet there is quite a discussion on peatlands and non-forests. These non forest areas are important in the conditions being discussed, but the text needs to be modified to make the connection clear.

The sentence on “We explore the failures, success stories, and criteria and indicators that have been developed for forest restoration.” does not begin to describe what is discussed in this review.   Writing out a few more sentences saying theoretical perspectives on land and forest restoration including hydrology, biodiversity, and carbon, and socioeconomic considerations are discussed, as well as policies.   Implementation of forest restoration and effects on soil microbial communities, livelihood benefits, etc, basically I suggest listing out the section titles of the paper here.  Including text throughout that better explains why these sections were chosen would be useful.  As written, some of the sections seem unexpected or out of place.  Section 5 on climate change mitigation seems like it could be a section within section 4, with additional subsections.  Why is section 5 its own section?  It is unclear why it stands alone.  L719-721 and L791-792 repeat the same idea, yet are citing different references.   There is some repetition in the text due to the organization of the sections; presenting a roadmap as to why these sections, and why this order would be a great help.  The introductory paragraphs in section 2 and 3 should be helpful to describe what is being presented in the rest of the section, and the other sections could benefit from one paragraph in the beginning to describe what is being presented in the section and how it relates to the paper.  The need for a comprehensive perspective (L212-213) is an important concept to explain why all these topics are needed but the sentence is buried in the paper.

Lines 989-995 seems like a paragraph that should be emphasized in conclusions, or an idea repeated in conclusions.

L1157-1159 lists out landscape types that should be mentioned in the introduction to make clear that what may be considered non forest in some ecosystems need to be considered forest in Indonesia.

Conclusions:  It seems like this section should be two paragraphs at least.  Please reconsider making modifications to make conclusions clear.

Minor items:

L014: has an extra “the” at end of sentence.

L1014: what does the term GoI stand for

Reviewer 2 Report

Congratulations to the authors on a very interesting and, in my opinion, exhausting review. Such a comprehensive approach can be useful for other countries that implement forest restoration programs.

The work is very interesting and describes the issue in all its complexity, multidisciplinarity and depth. The work is rather reminiscent of a very comprehensive research report. There would certainly be room for shortening, especially in the parts where the basic theoretical facts are described. e.g. lines 964 -970, but also elsewhere.

On the other hand, I find in the text parts that do not start from the reserse, but represent the analysis or synthesis of the authors, e.g. table 1 or figure 1 or lines 493-502, but also elsewhere in the text.

I see the biggest problem in the methodology. The reader does not know how the publication sources for the review were selected and does not know why such a text structure and also the order of the chapters was chosen. For example, I can imagine the use of policy analysis theory and a description of the policy cycle phases.

Reviewer 3 Report

General comments

The review titled “Tropical Forest Landscape Restoration in Indonesia” has been submitted to our appreciation. It is admitted that the deforestation is due to anthropogenic pressures (wood cutting, fire, overgrazing etc.) correlated with climatic variability. Hence, many efforts have been made at different levels to attenuate the effects of land degradation.

The manuscript is interesting to understand the efforts made on the restoration in the context of climate change and the failure of restoration projects. The review is so long and not attractive. I recommend to the authors to summarize deeply and present only the key messages in all sections of the review. Also, the graphical abstracts are more important in the review. The authors can add these figures that resume and would attract the readers. Therefore, the figures must be in high resolution. It is the same comments for your tables Those presented in this version are not satisfactory. I am surprising that any part of the review show your methodological approaches.

Specific comments

The author makes efforts searching the documents that talk about the restoration in the context of Indonesia.

Despite these efforts, based on the content of this manuscript, I have a number of concerns in the different sections of this review:

#Abstract

Please add some relevant keywords (two) to this study.

What are the key messages that the readers can keep after reading your abstract?

A paragraph in the abstract must show your methodological approach to carry out this review. Add a paragraph that explains your methodological approach.

#Introduction

Lines 69-72: add the references that support this statement.

Lines 82-84 according to you, what are the differences between afforestation and reforestation?

Line 389: where is the figure 1?

#other sections of the review

I will provide my appreciation on the next sections of the review after consideration of the comments and suggestions made above.

Round 2

Reviewer 2 Report

My comments and suggestions were accepted, so I recommend publishing the publication.

Reviewer 3 Report

My comments and suggestions made on the review have been taken account by the authors. I recommend acceptation of the review in Land.